# Reversal of nanomagnets by propagating magnons in ferrimagnetic yttrium iron garnet enabling nonvolatile magnon memory

Korbinian Baumgaertl[1] & Dirk Grundler [1,2] ✉

Despite the unprecedented downscaling of CMOS integrated circuits, memory-intensive machine learning and artificial intelligence applications are limited by data conversion between memory and processor. There is a challenging quest for novel approaches to overcome this so-called von Neumann bottleneck. Magnons are the quanta of spin waves. Their angular momentum enables power-efficient computation without charge flow. The conversion problem would be solved if spin wave amplitudes could be stored directly in a magnetic memory. Here, we report the reversal of ferromagnetic nanostripes by spin waves which propagate in an underlying spin-wave bus. Thereby, the charge-free angular momentum flow is stored after transmission over a macroscopic distance. We show that the spin waves can reverse large arrays of ferromagnetic stripes at a strikingly small power level. Combined with the already existing wave logic, our discovery is path-breaking for the new era of magnonics-based in-memory computation and beyond von Neumann computer architectures.

Charge-based electronics exploiting semiconductors and metals have led to the digital age and given rise to ever increasing data-processing power during the last decades. The significant development slows down however due to overheating of processors at increased clock frequencies as well as the data-transfer bottleneck between charge-based processors and a nonvolatile storage device like a magnetic hard disk[1,2]. So far, such a storage device exploits reprogrammable states in a materials class which differs from the semiconductors and therefore requires signal conversion. Waves for data processing like photons in optical computers already avoid charge flow and excessive Joule heating at high clock frequencies. However, even for optical computers the currently implemented von-Neumann device architecture still suffers from the signal-conversion bottleneck, particularly in case of memory-intensive machine learning applications. Magnonics which is based on charge-free angular momentum flow via spin waves (magnons), promises to overcome this dilemma. Making use of the bosonic quasiparticles[3], it offers low-power consuming

wave-based computing with microwave signals up to THz frequencies[4–6]. A technology platform for disruptive information technologies is foreseen if the so far elusive nanomagnet switching by propagating spin waves would be experimentally demonstrated. This switching process in magnonic circuits would be path-breaking for both nonvolatile storage of wave signals without conversion losses and in-memory computation[2,7,8]. Experiments presented in ref. [9] are encouraging in that spin waves in an antiferromagnet excited by incoherent spin torques reversed an integrated micromagnet. However, such spin waves can not be used for wave logic. In ref. [10], spin waves coherently excited in a magnon conduit by a microwave antenna induced the modification of magnetic domains. Here, the required power was however in the mW regime.

In this work, we follow a different approach. We excite spin waves in the ferrimagnetic insulator yttrium iron garnet $Y_3Fe_5O_{12}$ (YIG) by radiofrequency (RF) signals and observe the magnon-induced reversal of ferromagnetic nanostripes integrated on its

[1]Laboratory of Nanoscale Magnetic Materials and Magnonics, Institute of Materials (IMX), École Polytechnique Fédérale de Lausanne (EPFL), Lausanne, Switzerland. [2]Institute of Electrical and Micro Engineering (IEM), École Polytechnique Fédérale de Lausanne (EPFL), Lausanne, Switzerland. ✉e-mail: dirk.grundler@epfl.ch

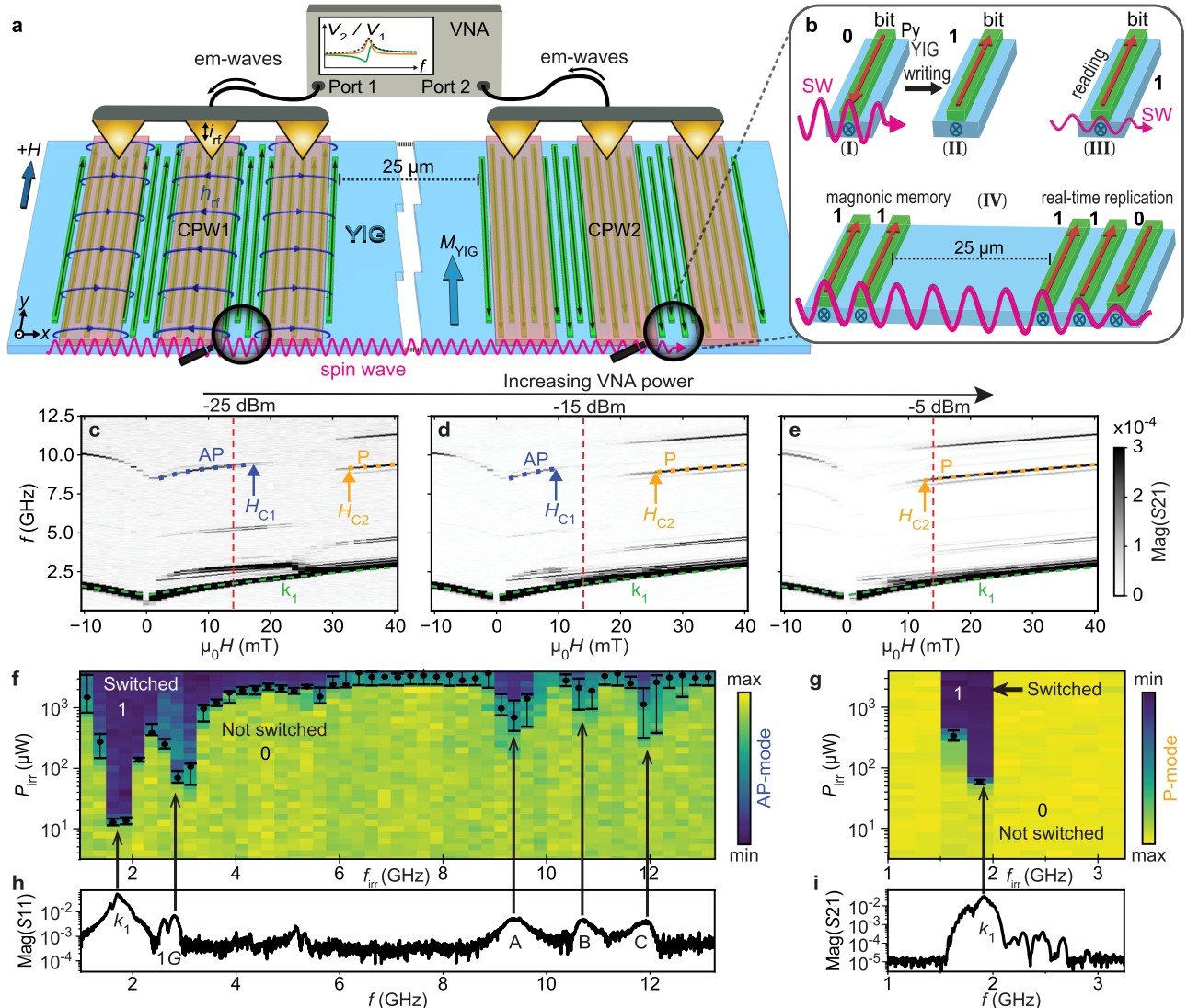

**Fig. 1 | Magnonic memory effect–Reading and writing of magnetic bits by spin waves. a** Two lattices of Py nanostripes (bistable magnetic bits) underneath coplanar waveguides (CPWs) on an insulating YIG film. **b** Depending on the spin-wave (SW) amplitude bit writing (I and II), reading (III) and data replication (IV) are achieved without charge flow. **c**–**e** Transmission signals Mag($S21$) are taken at three different power levels $P_{irr}$ by applying electromagnetic (em) waves via a vector network analyzer (VNA). Analyzing signal strengths of mode branches (blue and orange dashed lines), fields $H_{C1}$ and $H_{C2}$ are extracted reflecting bit reversal. The red dashed lines indicate +14 mT. Yield of bit writing (dark) underneath **f** CPW1 and **g** CPW2 by propagating SWs at $\mu_0 H_B = +14$ mT. The 50% transition power levels $P_{C1}$ and $P_{C2}$, respectively, are marked as black dots. The error bars indicate the 70% and 30% transitions (Methods). **h** Mag($S11$) at $P_{sens}$ and **i** Mag($S21$) taken at +14 mT. In **i**, bits at CPW1 and CPW2 were magnetized to state 1.

top surface (Fig. 1a). The shape anisotropy of the long and narrow permalloy (Py) stripes (green) allows for bistable magnetic states. Thereby, they represent magnetic bits with states 0 and 1, and enable the nonvolatile storage of information (Fig. 1b, I and II). The underlying YIG (blue) allows us to transmit coherent magnons over macroscopic distances as their decay lengths in thin YIG are on the few 10 to 100 $\mu$m length scale[11,12]. They are far larger than electron scattering lengths in semiconductors and metals. We note that samples similar to the one shown in Fig. 1a have been thoroughly investigated recently in view of the so-called grating coupler effect. Different groups explored nanostripe arrays on YIG in their saturated states for the emission and detection of short-wave magnons[13–16]. The magnon modes were transmitted over long distances[14] and imaged in real space by means of x-ray based magnetic microscopy[16]. However, the groups did not investigate if states of ferromagnetic nanostripes were different before and after spin-wave excitation at different power levels.

Here, we report experiments that we performed on YIG spin-wave buses (delay lines) with integrated grating couplers (Fig. 1a) and led to the discovery of irreversible switching of the nanostripes by propagating magnons. By combining power-dependent spin-wave spectroscopy and magnetic force microscopy we demonstrate the storage of magnon signals in the YIG-based magnonic circuit containing ferromagnetic stripes. Importantly, the magnon-induced reversal reported here is induced by magnons which are transported over a macroscopic distance ( > 20 $\mu$m). This goes beyond refs. [9,17–20] which considered distances on the nm length scale. Our finding is key for implementing data storage in wave-logic or neuromorphic computing architectures in magnonics[21]. Spin waves in YIG have already been shown to read out different magnetic states of integrated ferromagnetic stripes (Fig. 1b, III)[14,22]. Combined with our discovery, in-memory computing and a complete read-and-write process of nonvolatile magnetic bits via magnons in YIG circuits are now possible.

## Results

### Reversal of nanomagnets by spin-wave spectroscopy on YIG

We studied different YIG-based magnonic circuits with integrated ferromagnetic nanostripes which showed magnon-induced reversal of nanomagnets reported here. We explored ferromagnetic nanostripes with widths between 50 and 200 nm, lengths up to 27 $\mu$m, edge-to-edge separations down to 50 nm and different orientations with respect to spin-wave emitters. In this publication we focus on the magnonic circuit shown in Fig. 1a consisting of two nanostripe arrays which form grating couplers[23] on YIG underneath coplanar waveguides (CPWs). The insulating YIG with magnetization $M_{YIG}$ is 100 nm thick. The emitter (left) and detector (right) CPWs are connected to a vector network analyzer (VNA) which provides electromagnetic (em) waves at GHz frequencies at port 1. In such magnonic circuits information can be encoded in either the SW amplitudes or phases[4,24–26].

The magnetic component of the em wave, $h_{rf}$ (field lines sketched in Fig. 1a), is converted into propagating spin-wave (SW) signals at the CPW via the torque $-|\gamma|\mu_0 M_{YIG} \times h_{rf}$, where $\gamma$ is the gyromagnetic ratio and $\mu_0$ the vacuum permeability. We read out the SW signals both in reflection configuration at the emitter CPW1 and, after transmission through the YIG circuit, at the detector CPW2. The measured scattering parameters $S11$ and $S21$, respectively, are phase-resolved voltage signals induced by spin-precessional motion. The VNA offers a frequency resolution of 1 Hz and precisely adjusted power levels. In Fig. 1a, 25-$\mu$m- and 27-$\mu$m-long ferromagnetic nanostripes are integrated between CPWs and YIG (see Methods and Supplementary Fig. 1). These are bistable nanomagnets made from the ferromagnetic metal $Ni_{81}Fe_{19}$ (permalloy, Py). In earlier works such bistable nanomagnets were investigated in the linear regime and gave rise to the grating coupler (GC) effect[13,14,16,23]. Here, we report a nonlinear phenomenon in that GC stripes are reversed by spin waves which propagate through YIG. We show that thereby the storage of SW signals occur. To highlight the additional functionality of the ferromagnetic nanostripes we consider them to represent magnetic bits with either state 0 (magnetization vector $M_{Py}$ pointing along $-y-$ direction) or state 1 ($M_{Py}$ along $+y-$ direction). Our findings hence demonstrate that propagating SWs in YIG can not only sense (read) the state of such magnetic bits (Fig. 1b) as reported before[14,22], but also reverse (write) magnetic states at appropriate power levels which can be still in the linear SW excitation regime. The 100-nm-wide stripes considered here are arranged in separated periodic arrays with a common lattice constant $a = 200$ nm. For the read-out (sensing) of 0 and 1 states we apply a small power level $P_{sens}$ of only $-25$ dBm at port 1 (or smaller) and make use of the GC effect in $S11$ and $S21$ spectra. It has already been shown that the signal strength and precise frequency of GC modes depended decisively on the field direction and relative magnetization directions of nanostripes and YIG[14,15]. When the configuration of emitter and detector grating couplers were not symmetric the detection of emitted GC modes was not possible[27]. Using an in-plane magnetic field $H$ we reset the magnetic states of stripes to a well-defined initial state (erase cycle) before performing subsequent magnon-induced reversal experiments.

First, we show that the nanostripes in the magnonic circuit are switched from 0 to 1 by SWs exhibiting a certain amplitude. Figure 1c-e displays the magnitude Mag($S21$) of the frequency-dependent voltages at CPW2 as a function of field when measured for three different VNA power levels $P_{irr}$ at CPW1. Dark branches indicate SW propagation. Initially, all nanostripes were magnetized along the $-y-$ direction (erased) by applying $\mu_0 H = -90$ mT. Then $\mu_0 H$ was increased in 1 mT steps to the $+y-$ direction. For all powers $P_{irr}$, Mag($S21$) contained a prominent and continuous low-frequency branch (marked by green dashed lines) attributed to the $k_1$-mode of SWs excited by CPW1 directly in the YIG film[16]. The $k_1$-branch changed slope close to $\mu_0 H \simeq 0$ mT, indicating that the magnetization $M_{YIG}$ of the soft

magnetic YIG film was aligned with the external magnetic field for values of +1 mT and above. High frequency branches between roughly 7.5 and 12.5 GHz are attributed to the GC effect. They showed distinct transitions (jumps) in resonance frequencies. These frequency variations are known to indicate irreversible switching of Py nanostripes[14,15]. Following refs. [28,29], the jumps occur at the switching fields at which the magnetization direction of nanostripes reverse. We focus on the two prominent branches marked by blue and orange dashed lines, which, in the following, we denote antiparallel (AP) and parallel (P) mode, respectively (Supplementary Methods). $H_{C1}$ ($H_{C2}$) defined the critical field value for which the signal strength of the AP-mode (P-mode) decreased (increased) to 50% of its maximum value (see Methods and Supplementary Fig. 2). From Fig. 1 c to e, $H_{C1}$ and $H_{C2}$ diminished significantly with increasing $P_{irr}$ indicating a nonlinear process and the reversal of nanostripes. Before presenting the magnon-induced reversal by means of frequency-resolved switching-yield maps introduced below, it is instructive to present magnetic force microscopy (MFM) data. We perform MFM at remanence to demonstrate that the magnon-induced reversal of ferromagnetic nanostripes is nonvolatile.

### Evidence of nonvolatile states after magnon transport

Figure 2a displays an MFM image taken in the remnant configuration after we performed a broadband VNA measurement with $P_{irr} = -25$ dBm at $\mu_0 H_B = +14$ mT on the sample with initially reset magnetic stripes. The MFM data show nanostripes at CPW1 and CPW2 which are magnetized still along $-y-$ direction (state 0) and opposite to $M_{YIG}$, consistent with our definition of the AP branch (cf. Fig. 1c). Note that $\mu_0 H_B$ was intentionally below the evaluated onset field of quasistatic reversal of magnetic stripes (Supplementary Fig. 3) in order to reveal SW-induced reversal processes. Figure 2b shows the MFM data after applying an increased power level of $P_{irr} = -15$ dBm (31.6 $\mu$W) at $\mu_0 H_B = +14$ mT (cf. Fig. 1d). The remnant MFM image shows nanostripes which on the right side of the signal line of CPW1 are reversed to the $+y-$ direction (state 1). On the left side with $x < -1.1$ $\mu$m they are still magnetized along $-y-$ direction. Fallarino et al.[30] showed that SWs were excited nonreciprocally by a CPW for the same field geometry as used in our experiments. A large (small) SW amplitude was reported for the right (left) side of the signal line with $x > 0$ ($x < 0$). We attribute the spatially inhomogeneous switching of nanostripes to the non-uniform (nonreciprocal) amplitude distribution of SWs excited in YIG. Stripes at CPW2 were still magnetized in $-y-$ direction for $P_{irr} = -15$ dBm, with the exception of the two stripes on the left edge. For this experiment with an intermediate power level, a large number of stripes of the two gratings was magnetized in opposite directions. This magnetic disorder explained the vanishingly small signal amplitude for GC modes in Fig. 1d at 14 mT[27]. After a VNA measurement on again reset magnetic stripes with a high power of $P_{irr} = -5$ dBm (c.f. Fig. 1e) the MFM data show that the majority of stripes below the emitter CPW1 was switched (Fig. 2c). Below CPW2, about two thirds of the magnetic stripes were switched to 1 as well. Only the stripes farthest away from CPW1 showed still the initial state 0 (yellow arrow). The spatial variation agrees with a magnon-induced reversal process which requires a threshold amplitude of the SWs propagating under CPW2. As the amplitude of the propagating SWs decayed in positive $x$-direction, the required amplitude was not available for a large propagation distance from CPW1. The MFM-detected magnetic states hence represent a nonvolatile memory which records whether a certain threshold SW amplitude has locally been present (state 1) or not (state 0). Thus, wave-based computational results can be stored without signal conversion. The reestablished magnetic order underneath both CPWs by high VNA power in Fig. 2c explains the observation of the branch P at +14 mT in Fig. 1e. We note that the significant asymmetry of switched nanostripes observed in Fig. 2b and the subsequently presented experimental data allow us to exclude microwave-induced heating as the trigger for reversal. Heating in our continuous wave

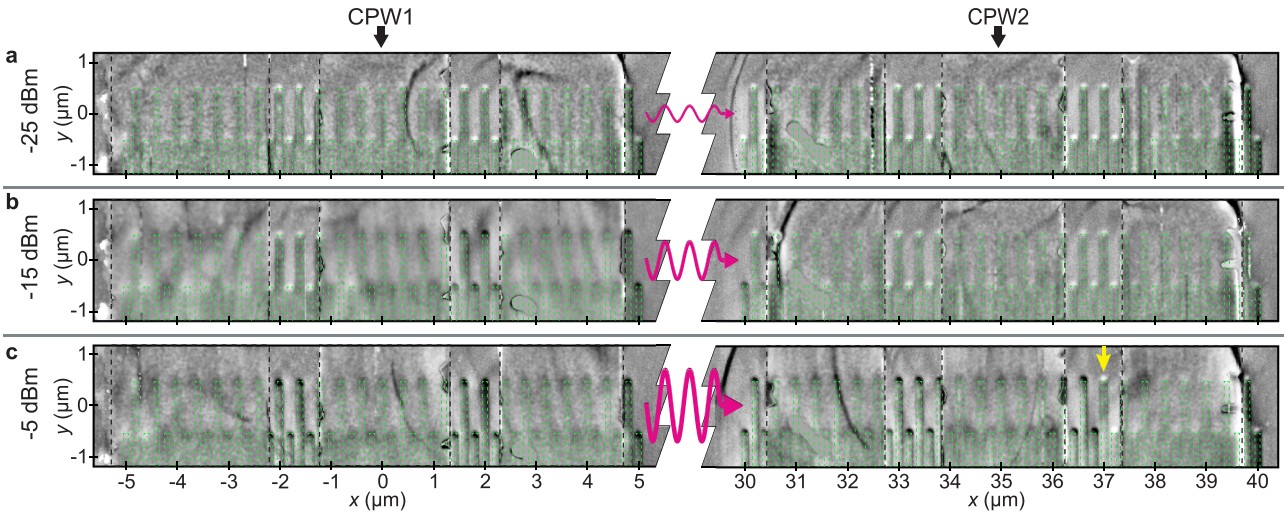

**Fig. 2 | Nonvolatile storage of spin-wave signals over macroscopic distance.** MFM measurements taken after the sample with erased bits (state 0, shown as white end) was irradiated at $\mu_0 H_B = +14$ mT with different powers **a** $P_{irr} = -25$ dBm, **b** $-15$ dBm, and **c** $-5$ dBm in the frequency range 0.1 GHz to 12.5 GHz. In this range, spin waves are excited in YIG. Magnetic bits (highlighted by broken lines) which are reversed appear with a black end. In **c**, beyond $x = 37\,\mu$m (marked by a yellow arrow) stripes remained magnetized in state 0 (white end), consistent with the decay of the SW amplitude when propagating away from the emitter CPW1.

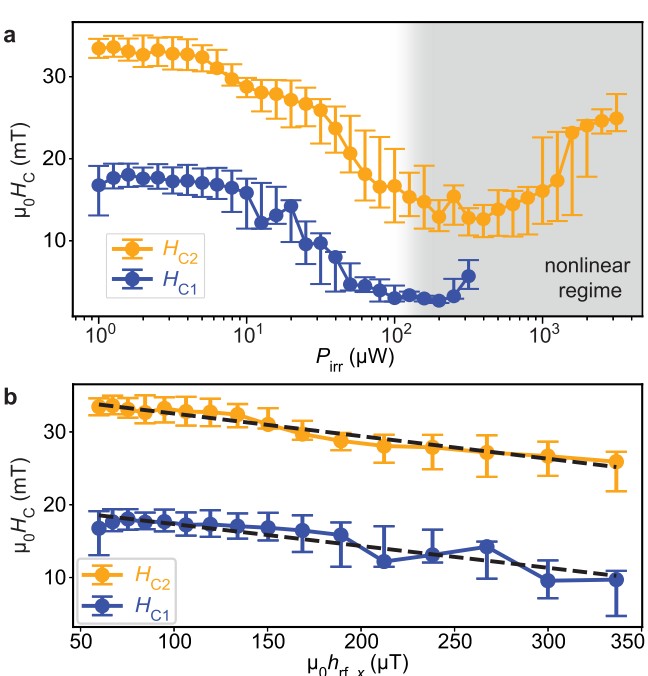

**Fig. 3 | Efficiency analysis of reversal field reduction by linear and nonlinear spin waves in YIG. a** Dependence of critical fields $H_{C1}$ (solid blue line) and $H_{C2}$ (solid orange line) extracted from VNA spectra on $P_{irr}$ applied in a broad frequency range from 0.1 GHz to 12.5 GHz. The power region attributed to nonlinear effects in the SW modes is shaded in light gray. **b** $H_{C1}$ and $H_{C2}$ as a function of the evaluated in-plane dynamic field amplitude $\mu_0 h_{rf,x}$ (rms-value) in the linear SW regime. The error bars represent the 30% and 70% switching field values. The straight lines are guides to the eye.

experiments would be most likely mirror-symmetric with respect to $x = 0$ of the emitter CPW. The direct electromagnetic cross talk between CPW1 and CPW2 is -50 dB ($10^{-5}$), i.e., vanishingly small.

## Power-dependent reversal field reduction

Figure 3a summarizes how critical fields $H_{C1}$ (blue line with dots) and $H_{C2}$ (orange line with dots) extracted from VNA data depended on the power used in VNA spectroscopy. We find that $H_{C1}$ ($H_{C2}$) decreases from 16.8 mT (33.4 mT) at $P_{irr} = 1\,\mu$W to 2.7 mT (12.9 mT) at 200 $\mu$W. This corresponds to a reduction of $H_{C1}$ by 84% and $H_{C2}$ by 61%. Surprisingly, for large $P_{irr}$ we find again an increase of $H_{C1}$ and $H_{C2}$. We attribute this observation to the onset of nonlinear SW scattering (Supplementary Fig. 4), which limits the peak amplitude of SWs emitted from CPW1[31]. For $P_{irr}$ above -9 dBm (shaded with gray in Fig. 3a) we found a red-shift of the $k_1$-mode resonance frequency (Supplementary Fig. 4), which is an indication of nonlinear SW excitation[32]. The reoccurring increase of critical fields is counter-intuitive if heating of Py nanostripes by high-power microwaves would be at the origin of the observed reversal phenomenon.

In Fig. 3 b we plot the experimentally extracted critical fields $H_{C1}$ and $H_{C2}$ (blue and orange symbols, respectively) versus the rms-value of the dynamical magnetic field component $\mu_0 h_{rf,x}$ of the radio-frequency field (RF) generated by the CPW. The amplitude is calculated via $\mu_0 h_{rf,x} = \mu_0 \sqrt{P_{irr}/(2Z_0 w_L^2)}$ [33] ($Z_0 = 50\,\Omega$ and $w_L = 2.1\,\mu$m, see Supplementary Fig. 1). The two dashed lines with negative slopes show that the nanostripes switch at smaller applied fields when $h_{rf,x}$ is increased. The top dashed line starts at values $\mu_0(h_{rf,x}, H_{C2}) = (x_1, y_1) = (0.06$ mT, 34 mT) and ends at $(x_2, y_2) = (0.34$ mT, 25 mT). Hence, an increase in RF amplitude of $\Delta(\mu_0 h_{rf,x}) = x_2 - x_1 = (0.34 - 0.06)$ mT $= 0.28$ mT leads to a switching field reduction by $\mu_0 \Delta(H_{C2}) = -(y_2 - y_1) = (34 - 25)$ mT $= 9$mT. The ratio $\Delta(H_{C2})/\Delta(h_{rf,x})$ amounts to 9 mT/0.28 mT $= 32.1$, that is, a small (positive) difference in $\mu_0 h_{rf,x}$ enables an about 30 times larger (negative) difference in $H_{C2}$.

## Magnon-induced reversal at low power level

To evaluate quantitatively the critical powers $P_C$ needed to reverse ferromagnetic nanostripes (switching-yield map), we excited SWs at well-defined frequencies $f_{irr}$, varied the excitation power $P_{irr}$ and explored after each power step whether the magnetic configuration of nanostripes changed (Supplementary Fig. 2). We defined $P_{C1}$ ($P_{C2}$) as the critical power value at which mode branches exhibited 50% of their maximum signal strength, indicating that a considerable number of Py nanostripes under CPW1 (CPW2) were switched; the results for an applied field $\mu_0 H_B = +14$ mT are summarized by black dots in Fig. 1f(g). Datasets for a separate sample are shown in Supplementary Fig. 5. We considered further $\mu_0 H_B < \mu_0 H_{C1}(-25 \quad$ dBm$) = 16.8$ mT and obtained consistent

diagrams (not shown). We find the minimum $P_{C1}$ of 12.6 $\mu$W when applying an RF signal to CPW1 in the frequency window $f_{irr}$ = 1.5 GHz to 1.75 GHz. This window includes the peak frequency of the $k_1$-mode (1.70 GHz) with a wavelength of 7.4 $\mu$m (Methods), as evidenced by the SW absorption spectrum Mag($S11$) (Fig. 1h). We note that this wavelength is much longer than both the width of stripes and the period $a$ of arrays. The wavelength is not commensurable with the stripe period $a$. The second lowest $P_{C1}$ of 68.9 $\mu$W occurs for $f_{irr}$ = 2.75 GHz to 3.0 GHz. This frequency range coincides with the frequency of the GC mode named $k_1 + 1G$. This mode exhibits a wavelength of 195 nm which is a value close to period $a$ (for mode identification see Supplementary Fig. 6 and ref. [16] for the P-mode).

The switching-yield map features three further dips at high frequency marked as A, B, and C. They occur at power values $P_{C1}$ of 0.69 mW, 1.91 mW, and 1.13 mW for $f_{irr}$ near 9.25 GHz, 10.75 GHz, and 11.75 GHz, respectively. These dips agree with eigenresonances of the Py stripes. At such frequencies, the concomitant reversal of Py stripes underneath CPW1 is attributed to the conventional microwave-assisted magnetization reversal (MAMR). MAMR of individual nanomagnets was pioneered in 2003[34] and then applied to mesoscopic Py magnets in e.g., refs. [28,29,33,35]. Those latter micromagnets were not exposed to a propagating spin wave. Instead, they were underneath a CPW and irradiated directly by an RF signal. The signal's frequency hit exactly the eigenfrequency of the Py. We measured the direct electromagnetic crosstalk between the CPWs. At CPW2, it amounted to -50 dB. In other words, the directly irradiated microwave power was five orders of magnitude lower and too small for MAMR at CPW2. The dips marked as A, B, and C in Fig. 1f that occur near 8 GHz will not be discussed in the following. The important features are the ones at low frequency in Fig. 1f, g. They go beyond MAMR in that (1) we excite spin precession in YIG and not in the Py nanostripes, (2) different spin-wave frequencies in the YIG induce switching of Py nanostripes and (3), nanostripes more than 25 micrometers away from the emitter CPW reverse (Fig. 1g).

The critical power applied to CPW1 for magnetic stripe reversal under CPW2 is $P_{C2}$ (Fig. 1g). We find $P_{C2}$ = 58.4 $\mu$W when exciting the $k_1$-mode in YIG (Fig. 1i). Its eigenfrequency was slightly blue-shifted compared to spectra taken for $P_{irr} < P_{C1}$ evidencing the concomitant magnetic stripe reversal by the $k_1$-mode at CPW1. Hence, a SW mode near 2 GHz can write states 1 at CPW1 and perform a real-time replication of these bit-states 1 about 25 $\mu$m away [Fig. 1b, panel IV]. Similar to ref. [22] we observed that reversed stripes changed the transmitted SW amplitude. Such a signal variation was recently functionalized in a neural network based on propagating magnons[21]. The maximum distance over which reversal is possible is determined by the intermediate damping as will be discussed in the following.

The minimum $P_{C2}$ inducing reversal at CPW2 is about 4.5 times larger than the minimum $P_{C1}$. We attribute this discrepancy to the damping of the propagating $k_1$ SW mode. Assuming $I(s) = I(x = 0) \exp(-s/l_d)$ and a decay length $l_d$ of 23 $\mu$m, the intensity $I$ of a SW would decay by 1/4.5 over a distance $x = s = 35 \mu$m, i.e., the center-to-center distance between CPW1 and CPW2. Based on the intrinsic Gilbert damping $\alpha_i = 9 \times 10^{-4}$ of the YIG film, we calculate a decay length of 99 $\mu$m for SWs with a wave vector $k_1$ in unpatterned YIG (Supplementary Fig. 7). It is reasonable to assume that the effective damping parameter in the magnonic circuit of Fig. 1 was larger due to additional magnon scattering and radiative losses[36]. We hence consider $l_d$ = 23 $\mu$m to be a reasonable value. The mode with wave vector $k_1 + 1G$ has a smaller velocity than the $k_1$-mode (Supplementary Fig. 7b) and thereby a further decreased decay length. This consideration explains why the maximum available power of the VNA was not sufficient for nanostripe reversal via the $(k_1 + 1G)$-mode underneath CPW2.

We now discuss the power transferred to the spin system of YIG when we observe the reversal of nanostripes. The microwave power applied by the VNA to the CPW on YIG is known to be mainly reflected.

The severe reflection occurs because of the large mismatch between velocities and wave vectors of the applied long-wavelength electromagnetic wave and the excited short-wavelength spin wave[23]. We use $P_{prec}$ to parametrize the microwave power transferred into magnetization precession in YIG. This power is much smaller than the input power $P_{irr}$ due to reflection. It is evaluated from $P_{prec}(f_{irr}) = P_{irr}(f_{irr}) \cdot [\text{Mag}(S11)(f = f_{irr})]^2$. Using the measured spectra Mag($S11$), we calculate corresponding power values $P_{C1,prec}$ = 36 nW and 3.4 nW for the magnetic stripe reversal by means of the $k_1$-mode and the short-wave magnon labeled $k_1 + 1G$, respectively. We note that the large amount of reflected power which is not used for the reversal is not lost. By means of a circulator it could be applied to further magnonic circuits.

## Discussion

The magnon-induced reversal of bistable ferromagnetic nanoelements via different propagating spin waves modes with wave vectors $k_1$ and $k_1 + 1G$ in YIG is a key asset when aiming at in-memory computation and beyond von-Neumann device architectures based on multi-frequency magnonic circuits. Further studies are needed to understand the microscopic mechanism behind the magnon-induced reversal reported here and thereby to predict the theoretical limit of minimum power consumption. But already at this stage several considerations can be made. CPWs and grating couplers emit spin waves at specifically designed wave vectors and due to the spin wave dispersion relation in YIG a specific frequency band is thereby realized. Importantly we observe low-power magnon-induced reversal for both a pure CPW mode ($k_1$-mode) and a GC mode ($k_1 + 1G$). The nature of the mode does not seem to play a decisive role. The observed frequency selectivity is so far attributed to the microwave-to-magnon transducer used and not the reversal process itself.

A decade ago, Slonczewski pointed out that magnons in a ferrimagnet/ferromagnet hybrid structure have the potential to provide spin-transfer torque for nanomagnet reversal more efficiently than charge flow[17]. His theoretical predictions reflected that for a given amount of energy more bosonic magnons than fermionic electrons could be generated. If transferred to an adjacent nanomagnet, each of these magnons provided an angular momentum of $\hbar$ for reversal. Beyond this so-called magnon torque, spin precessional motion in YIG was found to inject a spin-polarized current $\mathbf{j}_s \propto \mathbf{M}_{YIG} \times d\mathbf{M}_{YIG}/dt$ into Py[37]. The spin current $\mathbf{j}_s$ displayed a nonreciprocal behavior[38] and its amplitude increased with the SW wave vector in YIG[38,39]. The spatial symmetry of our MFM data of reversed stripes is consistent with both the predicted magnon torque or $\mathbf{j}_s$. Furthermore, the $k_1$- and ($k_1 + 1G$) modes have indeed shown a characteristically different power level for reversal. One might hence assume that magnon torque[9,10,19,20] and/or spin current[18,38] could play a role for the switching of Py stripes. We speculate however that also the dipolar magnetic fields of propagating spin waves introduce torques on Py magnetic moments. As the frequencies of SW modes with wave vectors $k_1$ and $k_1 + 1G$ in YIG are well below the eigenfrequencies of the Py nanostripes (Fig. 1f and g and Supplementary Fig. 5), they do not fulfill the condition for resonant chiral absorption discussed theoretically in ref. [40]. A possible scenario might instead consist of SW-induced nucleation of a domain wall followed by its propagation due to the applied static magnetic field.

We now compare the power levels used here with the ones reported for the pioneering spin-transfer torque experiment in ref. [41] performed in applied fields $H$. In ref. [41], a directly injected electrical (el) current was used to switch one single Co nanomagnet of a small volume $(60 \times 130 \times 2.5)$ nm$^3$. A minimum power $P^*_{el} \approx 6 \mu$W was required. In our study, we calculated a minimum power in the YIG spin system of only $P_{C1,prec}$ = 3.4 nW when reversing a few 10 Py nanostripes of a much larger volume $(100 \times 26000 \times 20)$ nm$^3$ each. In our so-far non-optimized magnonic devices, the required power levels are also

orders of magnitude below the ones recently reported for domain-wall manipulation by SWs in Py stripes and the torques produced by incoherent magnons in a thin antiferromagnetic barrier[9,10]. Stimulated by ref. [17], we argue that the bosonic nature of magnons is in general advantageous. In our experiments, the magnons in YIG had frequencies $f$ around 2 GHz. They were hence of low energy $hf$ where $h$ is the Planck constant. Consequently, a small power can generate a large number of identical low-energy magnons in a short time which then produce torques for reversal via different scenarios outlined above. We expect further reduced critical power levels for magnetic bit reversal and smaller $P_{C,prec}$ by optimizing microwave-to-magnon transducers, modifying the shape or volume of nanomagnets and engineering the interface between magnetic bit and magnon conduit via e.g., an antiferromagnetic layer.

## Methods

### Sample fabrication

The samples were prepared on a 100 nm thick yttrium iron garnet (YIG) film, which was grown by liquid phase epitaxy and purchased from the Matesy GmbH in Jena, Germany. On top of the insulating YIG, a 20 nm thick metallic Py ($Ni_{81}Fe_{19}$) film was deposited via electron beam evaporation. The grating pattern was defined via electron beam lithography (EBL) using negative hydrogen silsesquioxane resist and transferred into the Py film by $Ar^+$ ion beam etching. The etching time was optimized to ensure minimal overetching into the YIG film. Coplanar waveguides were prepared by EBL lift-off processing using a PMMA/MMA double layer positive resist and evaporation of Ti/Cu (5 nm/110 nm).

### Spin wave spectroscopy with a vector network analyzer

SW spectroscopy measurements were conducted with a VNA (Keysight PNA N5222A) in a microwave probe station with integrated in-plane magnetic field control. The magnetic field was created by electromagnets and stabilized via a Hall-sensor controlled feedback loop. High-frequency coaxial cables with 50 Ω impedance and microwave probes were used to connect port 1 and port 2 of the VNA to CPW1 and CPW2, respectively. For all conducted measurements, microwave power $P_{irr}$ (or $P_{sens}$) was only applied to port 1, i.e., only the scattering parameters $S11$ and $S21$ were measured. The front panel jumpers of the PNA N5222A were adjusted at port 2 to provide increased forward ($S21$) dynamic range. For measurement of the microwave spectra at different power levels (cf. Fig. 1c–e and raw data for Fig. 3) the frequency was scanned from 10 MHz to 12.5 GHz in 3.1 MHz steps. An IF bandwidth of 1 kHz was used. For the switching-yield measurements (cf. Fig. 1f, g) $f_{irr}$ was scanned in 2.5 MHz steps. An IF bandwidth of 1 kHz and a dwell time of 50 ms was used. The magnitudes of the reflection and transmission scattering parameters Mag($S11$) and Mag($S21$) were calculated as the square root of the sum of the respective squared real and squared imaginary parts, after removing nonmagnetic background signals, which did not depend on the applied external magnetic field. Mag($S11$)$^2$ is the measure of the relative absorbed power and used to calculate the power $P_{prec} = P_{irr} \cdot$ Mag($S11$)$^2$ transferred from the CPW into the spin-precessional (prec) motion in the magnonic circuit. The most prominent SW excitation of CPW1[16] occurred at a wave vector $k_1 = 0.85$ rad $\mu m^{-1}$. The wavelength of the corresponding $k_1$ mode in YIG amounted to about 7.4 $\mu m$. The periodic array of Py nanostripes gave rise to a reciprocal lattice vector $1G$ and enabled the excitation of short-wave magnons by the grating coupler effect[23]. Its wave vector amounted to $k_{1G} = 2\pi/a = 2\pi/(0.2\ \mu m) = 31.4$ rad $\mu m^{-1}$. The absorption spectrum Mag($S11$) in Fig. 1h showed a double peak structure for the resonant feature labeled with 1G. We attributed the smaller peak on the low-frequency side of 1G to a grating coupler mode with $-k_1 + 1G$ and the larger high-frequency peak to $k_1 + 1G$. The mode $k_1 + 1G$ exhibited an absolute wave vector of $k_{k1+1G} = k_1 + k_{1G}$. Its wavelength amounted to $2\pi/k_{k1+1G} = 195$ nm under the stripe array[16]. This value was close to $a$

due to $k_1 \ll k_{1G}$. The GC mode $-k_1 + 1G$ corresponded to a wavelength of 206 nm under CPW1. Considering its larger absorption strength we assumed mode $k_1 + 1G$ to induce mainly the reversal of nanostripes in the text. Simulations are performed using MuMax3[42] considering ref. [43] (Supplementary Fig. 6). The determination of the decay length is presented in Supplementary Fig. 7 making use of refs. [44–46].

### Analysis of switching fields

To determine the power and frequency dependence of the discovered switching process, the following measurement routine was conducted: The stripes were first magnetized with $\mu_0H = -90$ mT (state 0) and then the field was increased slowly to a positive bias field of $\mu_0H_B = 14$ mT. This field value was below $\mu_0H_{C1}$ at -25 dBm, i.e., below the evaluated minimum field of the switching field distribution (Supplementary Fig. 3). We monitored Mag($S21$) with a low sensing power $P_{sens} = -25$ dBm. This power level was chosen such that $P_{sens}$ itself was insufficient to induce switching. While staying at $\mu_0H_B = 14$ mT, we applied $P_{irr}$ in a small frequency window $f_{irr}$ with a range of 0.25 GHz. $P_{irr}$ was increased from -30 dBm to 6 dBm in 1 dBm steps. After each power step, Mag($S21$) was measured with $P_{sens}$ in a frequency window from 7.5 GHz to 12.5 GHz. Thereby we covered the branches indicating the magnetic configuration of the stripes. After each such sequence the sample was reset to the initial state (bits were erased) for the next frequency $f_{irr}$. Supplementary Fig. 2a shows exemplarily Mag($S21$) measured as a function of $P_{irr}$ with 1.75 GHz ≤ $f_{irr}$ ≤ 2.0 GHz. With increasing $P_{irr}$ the AP-mode vanishes and the P-mode appears. We evaluate mode strengths as a function of $P_{irr}$ (Supplementary Fig. 2b) by integrating Mag($S21$) in the vicinity of the AP-mode (enclosed by blue dashed lines) and P-mode (enclosed by orange dashed lines). We define $P_{C1}$ ($P_{C2}$) as the critical power value for which the signal strength of the AP-mode (P-mode) is at 50% of its maximum value (Supplementary Fig. 2b), indicating that 50% of the Py nanostripes in the grating under CPW1 (CPW2) were switched. Further we extract the values for which 30% and 70% of the signal strength are reached as a measure of the switching power distribution (error bar). The measurements were repeated for different $f_{irr}$ altered from 1 GHz to 13.25 GHz in 0.25 GHz wide windows. We quantified the relative numbers of reversed nanostripes by integrating the signal strengths in Mag($S21$) spectra in the vicinity of the AP- and the P-mode (Methods). At the low power level of $P_{sens} = -25$ dBm we analyzed the quasistatic switching field ($SF$) distribution of Py nanostripes under CPW1 (Supplementary Fig. 3a, b) and found a full width half maximum $SF_{S11,f} - SF_{S11,i}$ of roughly 13 mT centered around a switching field value of ($SF_{S11,f} + SF_{S11,i}$)/2 = 21.5 mT (Supplementary Fig. 3c). With increasing $P_{irr}$ we observed a decrease in values $SF$ and a narrower distribution (Supplementary Fig. 3c) consistent with the writing of bits reported in the manuscript.

## Data availability

The datasets generated during the current study are available from the corresponding author on reasonable request. The data sets analyzed for the manuscript are available in the Zenodo repository, https://doi.org/10.5281/zenodo.7714181.

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

## Acknowledgments
We acknowledge discussions with Andrea Mucchietto, Shreyas Joglekar, Michal Mruczkiewicz, and Anna Fontcuberta i Morral. The research was supported by the SNSF via grant number 163016 (D.G.).

## Author contributions
D.G. and K.B. planned the experiments and designed the samples. K.B. prepared the samples and performed the experiments. K.B. and D.G. analyzed and interpreted the data. K.B. and D.G. wrote the manuscript.

## Competing interests
The authors declare that they have no competing interests.
