## [Peer Review File · Nature Communications]

Reviewers' Comments:

Reviewer #1:

Remarks to the Author:

In the present paper, the authors reported a microwave-spectroscopic study on permalloy (Py) arrays on YIG. The result shows that the microwave spectra change steeply by applying strong microwaves, which the authors attribute to the microwave-assisted switching of magnetization configuration of the Py arrays, suggesting that the spatial distribution of the spin wave amplitude can be read out by using a microwave that comes later. The observed phenomenon can be applied to recording and reading of spin-wave interference patterns on magnetic films. The authors are aiming to develop it into non-Von Neumann computation.

What the paper provides is the proposal of new devices, not a finding of new phenomenon or new material; the authors utilize known physical mechanism (spin wave interference and microwave-assisted magnetization reversal) to propose a new device concept, with which the microwave spectrum is associated with the interference pattern of previously input spin standing waves in the device (hysteresis behavior). To evaluate the paper, therefore, one needs to be well informed of the advantage of the device based on the behavior. The present manuscript lacks this information, and I am afraid I cannot evaluate its novelty. If the authors resubmit the paper, please elaborate this point.

(additional comments)

1. There are known methods for recording spin wave special patterns: MOKE, local ISHE (embeddable in a device), local TMR (embeddable in a device), heat imaging (for standing wave only), BLS etc. Could you elaborate how to make use of the present all-magnon device? It would be helpful if you could give us an example of a concrete idea.

2. In the abstract "We attribute the high efficiency to the bosonic nature of magnons": This issue is not discussed in the main text. Why can the bosonic nature significantly improve the efficiency in the device?

3. In the introduction part: An introduction to spin wave computation from the view point of the non-Von Neumann computation is needed.

4. Quantitative discussion on the magnetization reversal mechanism is necessary.

Reviewer #2:

Remarks to the Author:

The authors present interesting experimental results for a magnonic device that appears able to control and sense the alignment of magnetic orientation encoded in Py bits using spin waves generated in a YIG underlayer. The results are interesting primarily because of the apparent small amplitude, low power, driving fields required to achieve bit reversal. The authors suggest that their scheme may have application for low power read/write operations for information storage using r.f. electromagnetic driving fields.

The geometry for the experiments is a coplanar transmission and coplanar detection scheme for the spinwaves, patterned on top of a YIG film that is decorated by an array of Py nanoelements. Dimensions for the Py element width and periodicity appear to be designed to be roughly commensurate with the spin wave wavelength excited in the YIG film, although this design aspect is not discussed explicitly.

Sensing appears possible through a dependence on the bit configuration observed in the spin wave spectra as a difference in frequency between parallel and antiparallel alignment of the bit magnetisations to the external field and YIG magnetisation. Writing appears possible through magnetic force microscopy observation of reversed bits for sufficiently large amplitude r.f. driving.

Notable features were observed with identification of thresholds of driving field amplitudes (not power) for sporadic Py reversal, thresholds for complete Py reversal, and ability to correlate spin wave frequency shifts with the resulting Py configurations. Additionally, there appeared to be evidence for bit reversal under each antenna, with reversal under the detection antenna occurring at a slightly different frequency corresponding to a grating generated wavelength. Further observations are: non-uniform reversal, which seems to be attributed to the asymmetric propagation characteristics of spin waves generated by the antenna; and evidence for reversal at higher frequencies associated with microwave assisted reversal in the Py.

The results are intriguing, and the paper focusses on the device operation characteristics while deferring much of the interesting physics explored to the references. As such there are many questions that appear unaddressed. Most notably,

1. As noted in the Supplemental Material, the interface needs to be better understood. There appears to be no evidence of strong exchange coupling between the YIG and Py, which is curious as these appear to have been fabricated without a spacer layer separation. However one wonders how exactly spin current generated torques on the Py are conveyed across the interface with YIG. There is little discussion of the mechanism enabling such apparently small driving field enabled reversal in the manuscript, which is unfortunate as this seems of primary importance for the observations.

2. The geometric design of the Py array periodicity with respect to spin wave wavelengths is not explained. Also, the Py element volumes with respect to the expected spin current densities necessary for element reversal are not discussed. In this regards one wonders about details of exactly how reversal is achieved: i.e. micromagnetic simulations might give some insight into the magnetic mechanisms involved under spin torques and how these are affected by the Py element shape.

3. Further in this regards, the wave length and lattice periodicity do not appear to be entirely commensurate. What might be the consequences of this, or are there any given the decay length of spin waves and occurrence of scattering processes?

4. It is claimed in the Supplemental information around line 35 that heat assisted switching is ruled out by observed frequency dependences. This is not entirely clear. Can the authors elaborate?

5. Finally, as a reader, I find the use of very many acronyms distracting and difficult to follow. The paper itself is dense and very concise, and it was a little difficult to find the thread of logic holding it together (although it exists). It would help considerably to be a bit more explanatory in the main text, and perhaps make more effective use of the Supplemental Information for embedding details and complications.

In summary, there is much that is very interesting in this work. The style of presentation is perhaps the weakest point. I am also concerned that the relatively low emphasis on new physics, with instead a high focus on device characteristics, may make the paper in its present form more suitable for a specialized journal.

Reviewer #3:

Remarks to the Author:

This paper reports the reading, writing, and real-time replication of magnetic bits by the spin waves. However, the conclusions and experiments should be further deliberated. I cannot recommend the publication of this paper in Nature Communications, due to the following critical reasons. The manuscript seems more suitable for some physical journals like PRB.

1. The authors emphasize too much on the low power consumption of their devices. The magnetic fields are still needed in the demonstration. They are suggested not to oversell it.
2. The manuscript claims that they solve the so-called von Neumann bottleneck or the problem of Moore's law. However, the size of the devices is still too large and seems difficult to be polished.

The authors should compare with more results in other electronic devices before drawing such a conclusion. The statement is not rigorous.

3. What is the non-marked high-frequency mod, like 5 GHz and 11 GHz? Why are there no resonance signals between AP and P states in Fig1 c-e?

4. The authors should discuss more the heat effect on the magnetization switching by injected such a high-power microwave. Should the change of the H_c and the magnetization switching result from the thermal effect?

5. The MFM results are not very clear. The authors seem to determine the switching by the edge of the nanostrips? How is the repeatability of the experiments, especially the MFM?

6. Is it possible that the magnetization switching by the spin wave is partial? How is the stability of the magnetization switching after the injected microwave? Could the switching be confirmed by other methods?

7. It seems that the operation windows for microwave power in writing, reading, or replicating programs are narrow, which may limit its potential in applications. This should be more discussed and also tell the readers how to improve it.

8. The authors claim that the advantage of the device is low power consumption in the conclusion part, which should be compared with more different memory or computing devices such as RRAM, MRAM, PCM, not just with the spin wave-induced domain wall motions.

In conclusion, the manuscript could not be accepted currently.

Detailed Response to REVIEWER COMMENTS

Our responses are in italics and green color.

Reviewer #1 (Remarks to the Author):

In the present paper, the authors reported a microwave-spectroscopic study on permalloy (Py) arrays on YIG. The result shows that the microwave spectra change steeply by applying strong microwaves, which the authors attribute to the microwave-assisted switching of magnetization configuration of the Py arrays, suggesting that the spatial distribution of the spin wave amplitude can be read out by using a microwave that comes later. The observed phenomenon can be applied to recording and reading of spin-wave interference patterns on magnetic films. The authors are aiming to develop it into non-Von Neumann computation.

What the paper provides is the proposal of new devices, not a finding of new phenomenon or new material; the authors utilize known physical mechanism (spin wave interference and microwave-assisted magnetization reversal) to propose a new device concept, with which the microwave spectrum is associated with the interference pattern of previously input spin standing waves in the device (hysteresis behavior). To evaluate the paper, therefore, one needs to be well informed of the advantage of the device based on the behavior. The present manuscript lacks this information, and I am afraid I cannot evaluate its novelty. If the authors resubmit the paper, please elaborate this point.

Response: We thank the reviewer for the careful reading of the manuscript. We apologize that the format of the original manuscript confused the reviewer. We have rewritten the manuscript in order to highlight that the observed reversal of ferromagnetic nanomagnets on top of a magnon-guiding ferrimagnetic layer is new by itself. The discovered reversal of nanomagnets by spin waves propagating underneath them promises that a holy grail of magnonics can become real, i.e., charge-less in-memory computation.

(additional comments)

1. There are known methods for recording spin wave special patterns: MOKE, local ISHE (embeddable in a device), local TMR (embeddable in a device), heat imaging (for standing wave only), BLS etc. Could you elaborate how to make use of the present all-magnon device? It would be helpful if you could give us an example of a concrete idea.

Response: As highlighted above, the manuscript has been rephrased and now highlights our of nanomagnet switching by propagating SW clearer. Unlike spin wave imagine techniques (MOKE, local ISHE) mentioned by the reviewer the nanomagnets used represent a non-volatile memory. We show that they allow for the storage of information provided by SW signals. Importantly our approach is an “all-magnon” approach. The high-frequency response of the system reflects the magnetic states. Thereby the magnetization configuration can be read out by SWs of small amplitude (“all-magnon device”). The reviewer is right that also other established techniques for reading magnetic bits (e.g. GMR sensors) could be used, but such a readout is beyond the scope of the paper.

2. In the abstract “We attribute the high efficiency to the bosonic nature of magnons”: This issue is not discussed in the main text. Why can the bosonic nature significantly improve the efficiency in the device?

Response: We now refer in more detail to the original theoretical work done by J.C. Slonzewski. We added sentences to the introduction to inform the reader.

3. In the introduction part: An introduction to spin wave computation from the view point of the non-Von Neumann computation is needed.

Response: We thank the reviewer for this comment. We have rephrased the whole manuscript to highlight our discovery in a clear manner. Thereby we also rephrased the prospects offered for in-memory computation. We have added a reference on neuronal networks using spin waves. We expect that our work stimulates in particular more theoretical work on how non-von Neumann computation with spin waves and in-memory computation can advance information technologies.

4. Quantitative discussion on the magnetization reversal mechanism is necessary.

Response: The switching yield diagram in Fig. 1 f and g as well as data in Fig. 3 and Supplementary Figs. 2 to 5 provide quantitative descriptions concerning the reported magnon-induced reversal phenomenon.

Reviewer #2 (Remarks to the Author):

The authors present interesting experimental results for a magnonic device that appears able to control and sense the alignment of magnetic orientation encoded in Py bits using spin waves generated in a YIG underlayer. The results are interesting primarily because of the apparent small amplitude, low power, driving fields required to achieve bit reversal. The authors suggest that their scheme may have application for low power read/write operations for information storage using r.f. electromagnetic driving fields.

The geometry for the experiments is a coplanar transmission and coplanar detection scheme for the spinwaves, patterned on top of a YIG film that is decorated by an array of Py nanoelements. Dimensions for the Py element width and periodicity appear to be designed to be roughly commensurate with the spin wave wavelength excited in the YIG film, although this design aspect is not discussed explicitly.

*Response: We thank the reviewer for the careful reading of the manuscript and the positive statements about the contents. We have rephrased the manuscript and extended the introduction to describe in detail the layout of the sample. We now explain explicitly the different functionalities of the stripes, i.e., as a grating coupler and a storage device. We stress that the switching by propagating SWs at CPW2 was demonstrated at the k_1 wavevector. This wavevector is **not** commensurate with the grating coupler periodicity.*

*Importantly, the grating coupler modes were sensitive to the magnetization direction of the Py stripes (this has already been reported in e.g. Liu, C., Chen, J., Liu, T. et al. Nat. Commun. **9**, 738, 2018) and thus allowed for a magnon read-out of the magnetic state of the “memory” which we discovered.*

Sensing appears possible through a dependence on the bit configuration observed in the spin wave spectra as a difference in frequency between parallel and antiparallel alignment of the bit magnetisations to the external field and YIG magnetisation.

Response: This is correct. In the new version of the manuscript we have added references for the benefit of the reader.

Writing appears possible through magnetic force microscopy observation of reversed bits for sufficiently large amplitude r.f. driving.

Response: We apologize for the extremely compact version of the first manuscript. The editor has allowed us to expand the manuscript and we now better emphasize that MFM is used as a technique to convince the reader that the discovered “data writing” by spin waves creates nonvolatile state in the ferromagnetic stripe arrays, as needed for a magnetic memory or storage device. We use MFM to relate the observed changes in spin-wave spectra with a change in the magnetic states of ferromagnetic stripes. We have introduced a subtitle to stress the importance of nonvolatility and MFM.

Notable features were observed with identification of thresholds of driving field amplitudes (not power) for sporadic Py reversal, thresholds for complete Py reversal, and ability to correlate spin wave frequency shifts with the resulting Py configurations. Additionally, there appeared to be evidence for bit reversal under each antenna, with reversal under the detection antenna occurring at a slightly different frequency corresponding to a grating generated wavelength. Further observations are: non-uniform reversal, which seems to be attributed to the asymmetric propagation characteristics of spin waves generated by the antenna; and evidence for reversal at higher frequencies associated with microwave assisted reversal in the Py.

The results are intriguing, and the paper focusses on the device operation characteristics while deferring much of the interesting physics explored to the references. As such there are many questions that appear unaddressed. Most notably,

1. As noted in the Supplemental Material, the interface needs to be better understood. There appears to be no evidence of strong exchange coupling between the YIG and Py, which is curious as these appear to have been fabricated without a spacer layer separation. However one wonders how exactly spin current generated torques on the Py are conveyed across the interface with YIG. There is little discussion of the mechanism enabling such apparently small driving field enabled reversal in the manuscript, which is unfortunate as this seems of primary importance for the observations.

Response: We agree with the referee. The important message at this point is that the spin waves stimulate switching even in the absence of strong exchange coupling. We have rewritten the introduction and discussion sections to better outline what is already known and what still needs to be investigated. We stress that further experimental and theoretical efforts are needed to understand the microscopic mechanism behind the power-efficient reversal.

2. The geometric design of the Py array periodicity with respect to spin wave wavelengths is not explained. Also, the Py element volumes with respect to the expected spin current densities necessary for element reversal are not discussed. In this regards one wonders about details of exactly how reversal is achieved: i.e. micromagnetic simulations might give some insight into the magnetic mechanisms involved under spin torques and how these are affected by the Py element shape.

Response: We have rephrased the introduction to explain the rationale behind the sample layout. We note that we have investigated several samples with different stripe arrays concerning width, period, edge-to-edge separation and orientation. In the paper we focus on the main discovery and the

sample(s) which displays the magnon-induced reversal in a very clear manner in the spectra and the MFM data. We note that the the switching by propagating SWs at CPW2 was demonstrated at the k_1 wavevector, which is not commensurate with the grating coupler periodicity. A similar switching at k_1 wavevector was also observed for nanomagnets which we arranged in a zig-zag pattern and tilted with respect to the CPWs. We agree with the reviewer that a detailed understanding of the reversal mechanism is important. We are extremely confident that our findings will spark further experimental and theoretical studies in this direction.

We simulated spin wave assisted switching in Mumax3 using the inbuilt inter-region exchange (AIP Advances 4, 107133 (2014)) with scaling factors $S=1$ and $S=0$. In our simulations, the reversal of a 20 nm thick and 100 nm wide Py nanomagnet on a 100 nm YIG film at a bias field of 14 mT was only achieved at large RF driving amplitudes (above 1 mT) at which the YIG resonance was driven into the nonlinear regime. We speculate that the low-power switching observed in our experiments is mediated by spin currents flowing between the YIG film to the Py layer. Spin currents due to magnetization precession are not considered in the framework of Mumax3.10. While we agree with the reviewer that micromagnetic simulations might be helpful, the computational treatment of interface effects is not trivial and should be the focus of a separate study.

3. Further in this regards, the wavelength and lattice periodicity do not appear to be entirely commensurate. What might be the consequences of this, or are there any given the decay length of spin waves and occurrence of scattering processes?

Response: The reversal of stripes under CPW2 is observed for the k_1 mode which is not commensurate with the periodicity of the grating. Reversal underneath CPW1 is observed for the k_1 mode and also for the k_1+1G mode which is almost commensurate. We speculate that the reversal is independent of the wavelength. This would agree with the paper of J.C. Slonczewski who did not consider any specific wavelength when providing his claim concerning the gain in torque (and reversal efficiency) by bosonic magnons compared to fermionic electrons.

4. It is claimed in the Supplemental information around line 35 that heat assisted switching is ruled out by observed frequency dependences. This is not entirely clear. Can the authors elaborate?

Response: The data in Fig. 1 f show that the required power for switching crucially depends on the irradiation frequency. The dips in Fig. 1f match the frequency of spin wave modes shown in Fig. 1h. At the k_1 spin wave resonance we find a P_{C1} of only 12.6 μW , while in regions without SW excitation P_{C1} is above 1 mW (which is larger by a factor of about 100). This pronounced frequency dependence of P_{C1} cannot be explained by Ohmic heating of the CPW. The Ohmic losses of a conductor should gradually increase with frequency. Due to the low Curie temperature of YIG (~ 560 K) its saturation magnetization is very sensitive to heating and a temperature increase should lead to a decrease of resonance frequencies. As shown in the supplementary Fig. 4, the k_1 mode resonance frequency starts to decrease for an irradiation power above -9 dBm (126 μW). The switching however was observed at a much smaller power of -19 dBm (12.6 μW).

5. Finally, as a reader, I find the use of very many acronyms distracting and difficult to follow. The paper itself is dense and very concise, and it was a little difficult to find the thread of logic holding it together (although it exists). It would help considerably to be a bit more explanatory in the main text, and perhaps make more effective use of the Supplemental Information for embedding details and complications.

Response: We have rephrased the manuscript and avoided acronyms at several instances.

In summary, there is much that is very interesting in this work. The style of presentation is perhaps the weakest point. I am also concerned that the relatively low emphasis on new physics, with instead a high focus on device characteristics, may make the paper in its present form more suitable for a specialized journal.

Response: Stimulated by the comments of reviewer and the editor we have rephrased and expanded the manuscript.

Reviewer #3 (Remarks to the Author):

This paper reports the reading, writing, and real-time replication of magnetic bits by the spin waves. However, the conclusions and experiments should be further deliberated. I cannot recommend the publication of this paper in Nature Communications, due to the following critical reasons. The manuscript seems more suitable for some physical journals like PRB.

1. The authors emphasize too much on the low power consumption of their devices. The magnetic fields are still needed in the demonstration. They are suggested not to oversell it.

Response: We thank the referee for the careful reading and detailed comments. We do not want to oversell. We have investigated several (other) samples concerning the magnon-induced switching and observe that the supporting bias field depends e.g. on the exact width of the nanostripes. This tells us that our devices are still non-optimized for reporting the minimum power for magnon-induced switching. Still we highlight that in the pioneering experiment on electrically driven STT in "equally" non-optimized nanopillars (which we cited in the original manuscript) the (large) electrical power for reversal of 6 microWatts was extracted at a finite field (and not at zero field). Thereby we considered a similar experimental setting for the comparison mentioned in the manuscript. We have expanded the discussion to make the known and open aspects more clear for the readers.

2. The manuscript claims that they solve the so-called von Neumann bottleneck or the problem of Moore's law. However, the size of the devices is still too large and seems difficult to be polished. The authors should compare with more results in other electronic devices before drawing such a conclusion. The statement is not rigorous.

Response: The von-Neumann bottleneck is very general and not related to a feature size. It is a consequence of the architecture proposed by von Neumann about 80 years ago, i.e., the physical separation of processor and storage medium. So far, different materials classes are used for the two components. Consequently the nanostructured processor and the nanostructured storage medium are in different macroscopic housings, connected by macroscopic wiring. These macroscopic components become obsolete when one can achieve efficient in-memory computation directly on the same chip. In addition, wavelogic offers new opportunities. In conclusion new design rules can be expected beyond Moore's law always encouraging further miniaturization as mentioned by the reviewer.

3. What is the non-marked high-frequency mode, like 5 GHz and 11 GHz? Why are there no resonance signals between AP and P states in Fig1 c-e?

Response: The periodic Py grating acts as a grating coupler (GC). By comparison with theoretical calculations the mode at 5 GHz can be attributed to the 1G+PSSW1 mode, and the mode at 11 GHz to the 2G+PSSW1 mode. PSSW stands for perpendicular standing spin wave confined between the top and bottom surface. A detailed discussion of the GC modes of the presented sample in the parallelly magnetized state is published in Nano Lett. 2020, 20, 10, 7281–7286.

We further observe that in the antiparallel state the GC modes are downshifted in frequency by a few hundred MHz, which agrees with our simulation shown in Supplementary Fig. 6 and was previously reported e.g. in Nat. Commun. 9, 738 (2018). The absence of GC modes in the intermediate region between AP and P state measured in transmission is attributed to a detuning between emitter (CPW1) and receiver (CPW2). As confirmed by MFM measurements (Fig. 2b) in the intermediate state, the emitting Py nanostructures below CPW1 were already reversed, while the nanostructures at CPW2 were still magnetized antiparallel to the external field. The emitting and detecting GCs are not in the identical state and detection is suppressed.

4. The authors should discuss more the heat effect on the magnetization switching by injected such a high-power microwave. Should the change of the H_c and the magnetization switching result from the thermal effect?

Response: There are several indications that heat effects don't play an important role.

1. The required power for switching crucially depended on the irradiation frequency as displayed in Fig. 1f. The dips in Fig. 1f match the frequency of spin wave modes shown in Fig. 1h. At the k_1 spin wave resonance we find a P_{C1} of only 12.6 μ W, while in regions without SW excitation P_{C1} is above 1 mW (e.g. by a factor ~ 100 larger). This pronounced frequency dependence of P_{C1} cannot be explained by Ohmic heating of the CPW. The Ohmic loss of a conductor should gradually increase with frequency.

2. Due to the low Curie temperature of YIG (~ 560 K) its saturation magnetization is very sensitive to heating and a temperature increase should lead to a decrease of resonance frequencies. As shown in the extended data Fig. 4, the k_1 mode resonance frequency starts to decrease for an irradiation power above -9 dBm (126 μ W). The quantitative analysis of the power dependence of H_c in Fig. 3b was done in the linear regime below -9 dBm, where significant heating can be excluded. As seen in Fig. 3a, for larger powers where heating might be significant, H_c was even increasing.

5. The MFM results are not very clear. The authors seem to determine the switching by the edge of the nanostructures? How is the repeatability of the experiments, especially the MFM?

Response: We have improved Fig. 2 stimulated by the comment of the reviewer. The correlation between nanostructure magnetization and SW mode shifts in grating couplers is well established in literature and we now provide relevant references. The MFM measurements were only done as an additional confirmation that reversed states are nonvolatile. The MFM measurements couldn't be performed in situ in the VNA probe station. The un- and remounting of the sample led to scratching of the CPWs and an additional uncertainty concerning the exact direction of the applied magnetic field. Therefore, we performed MFM measurements for selected fields and powers. We measured different stripe ends, i.e., the stripe ends on the right side of the first array and on the left side of the second array. The results were consistent. The repeatability of the experiments was checked on the reported samples and further ones via the modified spin-wave spectra in VNA measurements. Supplementary Fig. 5 shows that for a second nominally identical sample the measured switching-

yield diagram agreed well with the results presented in the main text. Further, all data points for H_C in Fig. 3a were measured two times, the average difference between both measurements was below 1 mT.

6. Is it possible that the magnetization switching by the spin wave is partial? How is the stability of the magnetization switching after the injected microwave? Could the switching be confirmed by other methods?

Response: The data in Supplementary Fig. 2b show that mode transitions are not abrupt but gradual. We assume this is primarily due to the inhomogeneous spin wave amplitudes below the CPW and a switching field distribution. The stochastic effects and small differences in the coercivity of each individual stripe due to geometrical imperfections are expected to add to the broadening. The MFM data were taken at zero field after unmounting the sample from the VNA station. These MFM data showed stripes which were reversed to a nonvolatile magnetic state. In future studies we will focus on time resolved techniques to address the questions by the reviewer.

7. It seems that the operation windows for microwave power in writing, reading, or replicating programs are narrow, which may limit its potential in applications. This should be more discussed and also tell the readers how to improve it.

Response: CPWs and grating couplers emit spin waves at prominent wave vectors and due to the spin wave dispersion relation in YIG in specific frequency bands. Importantly we observe low-power magnon-induced reversal for both a pure CPW mode (k_1 mode) and a GC mode. The nature of the mode does not seem to play a decisive role. The observed frequency selectivity is so far attributed to the microwave-to-magnon transducer and not the reversal process itself. Further experiments would be needed.

8. The authors claim that the advantage of the device is low power consumption in the conclusion part, which should be compared with more different memory or computing devices such as RRAM, MRAM, PCM, not just with the spin wave-induced domain wall motions.

Response: In the revised version we rephrased the introduction and thereby motivate why we use the charge-based spin-transfer torque as a reference. This is motivated by the paper of J.C. Slonzewski which we now discuss in the introduction. We note that so far the design rules for an optimized magnonic memory is not yet clear, and the theory behind the switching via coherent magnons has not yet been created. Considering these missing aspects we prefer to stay on the grounds prepared by J.C. Slonzewski.

In conclusion, the manuscript could not be accepted currently.

Reviewers' Comments:

Reviewer #1:

Remarks to the Author:

The author clarified the purpose of this paper: finding a new phenomenon "spin-wave induced magnetization reversal." Then, I am afraid the phenomenon is not particularly new: there is known similar phenomenon such as microwave assisted magnetization reversal (MAMR), where the applied microwave induces magnetization dynamics that promotes field-induced magnetization reversal. Note that, in the present phenomena, the authors also applied magnetic fields, which means that the magnetization reversal was not realized by the spin wave alone. The only difference seems to be that MAMR uses magnetization dynamics confined in a small magnet, while this paper (spin-wave induced magnetization reversal) uses spin waves. I am afraid I could not call it a big step forward, and thus I cannot recommend publishing this manuscript in Nature communications.

(minor points)

1. The experimental demonstration of the magnetization reversal by spin waves claimed by the authors still lacks some essential experiments.

- The magnetization reversal of spatially asymmetric Py nanowires is attributed to surface spin waves, but MFM imaging does not measure the directional and frequency dependence of the magnetic field, which is necessary to prove that the magnetization reversal is due to magneto-static surface waves. Therefore, we cannot say that we have proved that the effect is due to the surface wave only by its spatial dependence. (The effect might be due to the shape of the sample, or the disturbance of the sample shape during microfabrication.)

- The effect of temperature has not been eliminated well. When the spin wave mode and the microwave frequency coincide, the heat generation due to microwave absorption becomes larger as well. This tendency can explain the frequency dependence of the threshold value seen in Fig. 1f,g.

- For the same reason as above, the frequency dependence of the threshold of the external reversal field is not evidence that spin waves are the origin of the switching.

- Why does the nonlinear magnon excitation region described in Fig. 3, which exceeds the threshold by a factor of 10, show almost the same switching field as the linear excitation region? When the microwave power exceeds the threshold by a factor of 10, the magnon phase should be disturbed by the strong nonlinearity. If coherent propagation of spin waves is important, as the authors claim, it is unlikely that the threshold is the same as in the linear excitation region.

2. Direct writing of memory by spin waves is hardly realized in a way that can be introduced into existing spin-wave computing technologies.

- The dependence of spin wave intensity or phase on the direction of magnetization reversal, which is essential for bit manipulation by spin waves, is not shown. We believe that the memory function as a bit recording has been hardly realized.

- The direction of the inversion is determined by the direction of the external magnetic field, and the spin wave only assists the inversion. Therefore, this is not an operational demonstration of a memory that should record spin wave information.

- In the case of utilizing spin waves for Neuromorphic computing, which is claimed by authors, what should be recorded are the weights of the nodes, which are continuous quantities, and it is difficult to say that these computers can be realized by recording only discretized 0,1, such as the direction of magnetization.

Reviewer #2:

Remarks to the Author:

The revised manuscript addresses well my two primary concerns: presentation style and appropriateness for publication in the present journal. I believe that the manuscript is now easier to follow in that the thread of argument is made more clear, and clearly highlights questions that require (and inspire) further work both theoretical and experimental concerning the fundamental mechanisms at work. I also find that the revised manuscript does concentrate on the scientific aspects of a problem made interesting for its potential application to technology. The scientific

challenge of understanding the physical mechanisms of magnon reversal across this interface are highlighted, and the paper presents important data that will help further this larger question. I recommend publication in its present form.

Reviewer #3:

Remarks to the Author:

The authors try to reply to the reviews' comments and questions. And they have made some improvements to the manuscript. However, I still do not recommend its publication in Nature Communications.

As the authors wrote in the introduction, the spin waves in their devices have already been published in their previous paper (Nano Letters 2021). What they have done in this paper is try to figure out the magnetic states under different rf power. The conclusions are still drawn by the spin-wave data. There seems no direct evidence observed. As they said, the MFM is only for support. More importantly, also figured out by another review, this paper focus on the properties of the devices, but the physical or the mechanism of the switching is not discussed. The RF could introduce many factors, like heat, strain, rf fields, spin currents... which may affect the magnetizations. From this concern and as well the innovation, this paper seems not to reach the level for publication in Nature Communications.

Besides, I am still concerned about the practicability of these proposed devices which is difficult to be achieved in the industries. The feature size is limited, the reliability is unclear and the operating margin is quite narrow. As I mentioned in the last report, they did not give a possible direction for improvements. In this way, it seems not to be of wide interest. The authors display the AP to P switching by the higher power of the rf signal and present its non-volatile. How could they switch back (P to AP states) again? Should the magnetic field also need to be reversed. What is the advantage of it compared with conventional field-induced switching? If the authors discussed the power consumption, the fields should also be taken into consideration. Finally, I could not find the results about the switching of the FIM (YIG) as the author claim as one of their main contributions.

Referee A (in Red Color):

“Then, I am afraid the phenomenon is not particularly new: there is known similar phenomenon such as microwave assisted magnetization reversal (MAMR)” [...] “The only difference seems to be that MAMR uses magnetization dynamics confined in a small magnet, while this paper (spin-wave induced magnetization reversal) uses spin waves.” –

Response: We have discussed microwave assisted magnetization reversal (MAMR) in our original manuscript. We have argued that MAMR achieved by direct radiofrequency (RF) signal irradiation does NOT allow to store the information given by the propagating spin waves which are foreseen for wave-logic applications in magnonics. In the revised version we have extended the discussion about the difference between MAMR and the magnon-induced reversal. We now write explicitly the following (in blue):

The switching-yield map features three further dips at high frequency marked as A, B, and C. They occur at relatively large power values P_{irr} of 0.69~mW, 1.91~mW and 1.13~mW for f_{irr} near 9.25~GHz, 10.75~GHz and 11.75~GHz, respectively. These dips agree with eigenresonances of the Py stripes. At such frequencies, the concomitant reversal of Py stripes underneath CPW1 is attributed to the conventional microwave-assisted magnetization reversal (MAMR). MAMR of individual nanomagnets was pioneered in 2003 [cite{Thirion2003}] and then applied to mesoscopic Py magnets in e.g. Refs. [cite{Nozaki2007, Topp2009, Nembach2007, PhysRevLett.99.207202}]. These micromagnets were not exposed to a propagating spin wave. Instead, they were irradiated directly by an RF signal. Its frequency hit exactly the eigenfrequency of the Py. We measured the direct electromagnetic crosstalk between the CPWs. At CPW2, it amounted to -50 dB. In other words, the directly irradiated microwave power was five (!) orders of magnitude lower and too small for MAMR at CPW2. The dips in Fig. \ref{Fig1}f that occur near 8 GHz will not be discussed in the following. The important features are the ones at low frequency in Fig. \ref{Fig1}f and g. They go beyond MAMR in that (1) we excite spin precession in YIG and not in the Py nanostructures, (2) different spin-wave frequencies in the YIG induce switching of Py nanostructures and (3), nanostructures more than 25 micrometres away from the emitter CPW reverse (Fig. \ref{Fig1}g).

“The magnetization reversal of spatially asymmetric Py nanowires is attributed to surface spin waves, but MFM imaging does not measure the directional and frequency dependence of the magnetic field, which is necessary to prove that the magnetization reversal is due to magneto-static surface waves.” –

Response: The nonreciprocity of surface spin waves has been studied extensively during the years in different samples [V.E. Demidov et al., Appl. Phys. Lett. 95, 112509 (2009); K. Sekiguchi et al., "Nonreciprocal emission of spin-wave packet in FeNi film", Appl. Phys. Lett. 97, 022508 (2010)]. Our group has reported the non-reciprocity in thin YIG in H. Yu et al., Sci. Rep. 4, Article number: 6848 (2014). **The striking asymmetry of MFM data with respect to the central coordinate of CPW1 in Fig. 2 (b) (reproduced below) shows a one-to-one correspondence with this established knowledge about non-reciprocal magneto-static surface waves.** Considering the reviewer’s comment, we have used an independent means recently for local probing of the non-reciprocity of the emitted spin waves and concomitant reversal. We performed spatially resolved measurements using micro-focus BLS (not shown). The results support the conclusions drawn in the present manuscript.

We noted in the original manuscript (and provided references) that the spin-wave spectra themselves show the reversal of stripes clearly, independent of the MFM data. We had mentioned this also in an earlier response: *Importantly, the grating coupler modes were sensitive to the magnetization direction of the Py stripes (this has already been reported in e.g. Liu, C., Chen, J., Liu, T. et al. Nat. Commun. 9, 738, 2018) and thus allowed for a magnon read-out of the magnetic state of the “memory” which we discovered.* To further support our view, we refer to A. Papp et al (Nature Commun. 12:6422 (2021)): “A spin-wave scatterer is a magnetic thin film with spatially non-uniform magnetic field acting on it: this magnetic-field distribution locally changes the dispersion relation of the wave, scatters (steers) the spin waves.” Correspondingly, the measured spin-wave spectra directly reflect the reversal of nanomagnets on YIG.

“The effect of temperature has not been eliminated well. When the spin wave mode and the microwave frequency coincide, the heat generation due to microwave absorption becomes larger as well. This tendency can explain the frequency dependence of the threshold value seen in Fig. 1f,g. [...] For the same reason as above, the frequency dependence of the threshold of the external reversal field is not evidence that spin waves are the origin of the switching.” –

Response: The resonance frequency of YIG itself is a **sensitive temperature sensor**. **We had written in the last response letter:** *Due to the low Curie temperature of YIG (~560 K) its saturation magnetization is very sensitive to heating and a temperature increase should lead to a decrease of resonance frequencies. As shown in the supplementary Fig. 4, the k_1 mode resonance frequency starts to continuously decrease for irradiation powers above -9 dBm (126 μ W). The switching however was observed at a much smaller power of -19 dBm (12.6 μ W). [...] As seen in Fig. 3a, for larger powers where heating might be significant, H_c was even increasing.*

Supplementary Fig. 4. | Onset power level of nonlinear spin-wave regime. Power dependence of resonance frequency f_{res} (red color) and full width half maximum (FWHM) of the k_1 mode resonance in Mag(S11) extracted by fitting with a Lorentzian function at $\mu_0 H = -10$ mT. Above $P_{\text{irr}} = -9$ dBm (indicated by dashed arrow) f_{res} starts to decrease significantly with increasing power. Around the same P_{irr} the FWHM starts to increase significantly. These two observations are attributed to the onset of the nonlinear regime of SW excitation.

Please note that at the frequencies near 2 and 3 GHz of modes k_1 and 1G (see the graph below) the P_y itself is not in resonance, is not heated directly following the reviewer's comment and switches nevertheless. Thus, the argument of the reviewer does not consider earlier arguments and does not apply to the reported phenomenon.

“Why does the nonlinear magnon excitation region described in Fig. 3, which exceeds the threshold by a factor of 10, show almost the same switching field as the linear excitation region? When the microwave power exceeds the threshold by a factor of 10, the magnon phase should be disturbed by the strong nonlinearity. If coherent propagation of spin waves is important, as the authors claim, it is unlikely that the threshold is the same as in the linear excitation region.”

Response: Here, the referee provides two statements about our manuscript which are NOT part of the paper. (1) **We do not stress that coherent magnons by themselves are important for the process. In contradiction to the claim of the referee we stated:** Recent experiments presented in Ref. [Wang2019b] are encouraging in that **incoherently** excited spin waves in an antiferromagnet reversed an integrated micromagnet.

(2) The threshold powers **are not found to be the same in the linear and nonlinear magnon excitation regimes** as shown in the figure 3 reproduced below. They vary in agreement with the expectation of the reviewer.

FIG. 3. Efficiency analysis of bit writing by linear and nonlinear spin waves. **a** Dependence of H_{C1} (solid blue line) and H_{C2} (solid orange line) on P_{irr} applied in a broad frequency range from 0.1 GHz to 12.5 GHz. The power region attributed to nonlinear effects in the SW modes is shaded in light gray. **b** H_{C1} and H_{C2} as a function of the evaluated in-plane dynamic field $\mu_0 h_{rf,x}$ in the linear SW regime. The error bars represent the 30 % and 70 % switching field values. The straight lines are guides to the eye.

“The dependence of spin wave intensity or phase on the direction of magnetization reversal, which is essential for bit manipulation by spin waves, is not shown. We believe that the memory function as a bit recording has been hardly realized.” –

Response: This statement of the reviewer contradicts results published in journal articles. A. Papp et al. showed by micromagnetic simulations in *Nanoscale neural network using non-linear spin-wave interference* (Nature Commun. 12:6422 (2021)): A spin-wave scatterer is a magnetic thin film with spatially non-uniform magnetic field acting on it: this magnetic-field distribution locally changes the dispersion relation of the wave, scatters (steers) the spin waves, creating an interference pattern. Consistently, we had stated in our manuscript: In Ref. [22] it was reported that reversed bits changed the transmitted SW amplitude. We have explicitly shown that a specific excitation power, i.e., a specific threshold amplitude of the spin waves, is needed to switch a nanostripe. Thereby, the outcome of a wavelogic operation which increases the SW amplitude by interference beyond the threshold value can be stored directly, even after 25 micrometres of propagation path.

[22] Qin, H., Hollaender, R. & Flajsman, L et al. Nanoscale magnonic Fabry-Perot resonator for low-loss spin-wave manipulation. Nat. Commun. 12, 2293 (2021).

“The direction of the inversion is determined by the direction of the external magnetic field, and the spin wave only assists the inversion. Therefore, this is not an operational demonstration of a memory that should record spin wave information.”

Response: We disagree and refer the referee to a commercial storage product where an additional signal is used to prepare the storage functionality. The Flash RAM uses a reset signal in that tunneling electrons change the gate's electronic charge in "a flash" (hence the name), clearing the cell of its contents so it can be rewritten (<https://www.computerworld.com/article/2550624/flash-memory.html>). A similar “flash” concept could be applied to a magnonic memory.

“In the case of utilizing spin waves for Neuromorphic computing, which is claimed by authors, what should be recorded are the weights of the nodes, which are continuous quantities, and it is difficult to say that these computers can be realized by recording only discretized 0,1, such as the direction of magnetization.” –

Response: A. Papp et al. (in *Nanoscale neural network using non-linear spin-wave interference*, Nature Commun. 12, 6422 (2021)) have shown how to perform **neuromorphic computing with nanomagnets offering 0 and 1 states using** interference of coherently driven **spin waves**. V.V. Kruglyak who is a pioneer in magnonics suggests the following: “[Spin-wave assisted switching is ...] of great use for in-memory computing and if training arrays of chiral magnonic resonators (e.g., like those shown in Fig. 8) as artificial neural networks.” (in “Chiral magnonic resonators: Rediscovering the basic magnetic chirality in magnonics”, Appl. Phys. Lett. 119, 200502 (2021))

Report C (in Red Color):

“As the authors wrote in the introduction, the spin waves in their devices have already been published in their previous paper (Nano Letters 2021).” –

Response: We do not discuss the properties of spin waves themselves. Instead we exploit the knowledge contained in Nano Letters 2021 and then explore a completely different phenomenon, i.e., spin waves change irreversibly magnetic states in ferromagnetic nanostripes. Our previous titles as well as the modified one reflect the different focus:

“Reversal of nanomagnets by propagating magnons in ferrimagnetic yttrium iron garnet enabling nonvolatile magnon memory”.

“The conclusions are still drawn by the spin-wave data. There seems no direct evidence observed. As they said, the MFM is only for support.”

Response: We do **NOT** find the statement “**only for support**” in our manuscript as suggested by the reviewer’s comment. Instead, we wrote: We use **MFM to demonstrate** that the reported magnon-induced reversal of ferromagnetic nanostripes leads to **non-volatile magnetic states**. We wrote: “The MFM-detected magnetic states hence represent a non-volatile memory which records whether a certain threshold SW amplitude has locally been present (state “1”) or not (state “0”). Thereby, wave-based computational results can be stored without signal conversion.” We had introduced a subtitle to **highlight the importance of MFM**. Correspondingly, in the earlier response letter we had written: *We use MFM to relate the observed changes in spin-wave spectra with a change in the magnetic states of ferromagnetic stripes. We have introduced a subtitle to stress the importance of nonvolatility and MFM.*

“More importantly, also figured out by another review, this paper focus on the properties of the devices, but the physical or the mechanism of the switching is not discussed.”

Response: **We had openly discussed different possible mechanisms**. We had written: “But already at this stage several considerations can be made. ...”. We expect that our paper and the data stimulate further theoretical work, like the one published in Ref. [Phys. Rev. Applied 15, 034089 (2021)]. We note that currently there is a controversy about the earlier published experimental work [Y. Wang et al., Science 366, 1125 (2019)] as either the magnon- or the conventional electron-mediated spin-transfer torque might have been relevant for the reported switching of magnetization adjacent to an antiferromagnet layer (see Phys. Rev. Applied 15, 034089 (2021)). New data on magnon-induced switching like ours which were obtained in a completely different experimental setup which enables wave-logic with storage will have a high impact. The data are urgently needed to develop a correct microscopic understanding.

“The RF could introduce many factors, like heat, strain, rf fields, spin currents... which may affect the magnetizations. From this concern and as well the innovation, this paper seems not to reach the level for publication in Nature Communications.”

Response: This is a broad collection of arguments. The reviewer might have overlooked that **we had addressed already many of the mentioned factors**: heat (see response above), rf fields (we switch remotely positioned nanomagnets, the cross talk is -50 dB, see above), spin currents (we mention this possibility explicitly in the discussion section). The argument about strain does not hold at all as we use Permalloy. Permalloy is known to avoid strain effects on magnetization. There is currently a scientific debate (see above) and our experiments contribute with an innovative approach in that for the first time we investigate magnon-induced switching in a frequency resolved manner. In addition, we report extremely low (and promising) power levels and show the switching for spin waves that propagate for 25 microns instead of a few nm.

“How could they switch back (P to AP states) again? Should the magnetic field also need to be reversed.”

Response: This would be one option. FLASH RAM exploits a “switch back” mechanism already (see above).

“The feature size is limited, the reliability is unclear and the operating margin is quite narrow. As I mentioned in the last report, they did not give a possible direction for improvements.”

Response: The reviewer might have overlooked our discussion. **We discussed these aspects in our previous version of the manuscript already:**

We expect further reduced critical power levels and smaller $P_{C,prec}$ by reducing the magnetic bits to the nanoscale and optimizing microwave-to-magnon transducers as well as the YIG/Py interface, respectively. Considering Ref. [17], the exact wavelength of the magnon might not be relevant and power-efficient reversal might be accessible over a broad frequency regime.

In the response letter we had written: *CPWs and grating couplers emit spin waves at prominent wave vectors and due to the spin wave dispersion relation in YIG in specific frequency bands. Importantly we observe low-power magnon-induced reversal for both a pure CPW mode (k_1 mode) and a GC mode. The nature of the mode does not seem to play a decisive role. The observed frequency selectivity is so far attributed to the microwave-to-magnon transducer and not the reversal process itself.*

Considering our discussion, there is no hint that the “operating margin is quite narrow” as the reviewer suggested. We have extended the discussion section, added the green paragraph and critically discussed a further mechanism based on dipolar interaction.

“What is the advantage of it compared with conventional field-induced switching?”

Response: We stated in the manuscript that following our experimental data the spin waves are 30 times more effective than the switching by a global magnetic field. There was also a long paragraph in the revised version by which advantages were explained. We reprint a screenshot of the paragraph here:

199 with our MFM data and the magnetic bit reversals initiated by k_1 and $k_1 + 1G$ modes, respectively. We now compare
200 the power efficiency reported here with the pioneering spin-transfer torque experiment in Ref. [16]. In Ref. [16] a
201 directly injected electrical (el) current was used and a minimum power $P_{el}^* \approx 6 \mu\text{W}$ was required to switch *one single*
202 Co nanomagnet of a small volume ($60 \times 130 \times 2.5$) nm^3 in applied fields H . In our study we evaluated a power of
203 $P_{C1,prec} = 3.4 \text{ nW}$ when reversing *a few 10* Py nanostructures of a much larger volume ($100 \times 26000 \times 20$) nm^3 *each* in
204 a so-far non-optimized magnonic device. Our power levels are also orders of magnitude below the recently reported
205 domain-wall motion by SWs and torques produced by incoherent magnons [9, 10]. Stimulated by Ref. [17] we argue
206 that the bosonic nature of magnons is key to understand the small power needed for magnet reversal in our studies.
207 The magnons in YIG had frequencies f around 2 GHz. They were hence of low energy hf . Consequently, a small
208 power is enough to generate a large number of identical low-energy magnons in a short time which then produce
209 the torque for magnetic bit reversal via a pure spin current \mathbf{j}_s . We expect further reduced critical power levels and
210 smaller $P_{C,prec}$ by reducing the magnetic bits to the nanoscale and optimizing microwave-to-magnon transducers as
211 well as the YIG/Py interface, respectively. The corresponding experiments will certainly contribute to a microscopic

The reviewer might have overlooked the relevant sentences and the paragraph.

“If the authors discussed the power consumption, the fields should also be taken into consideration.”

Response: In the earlier response we had written that our devices were not yet optimized: *We have investigated several (other) samples concerning the magnon-induced switching and observe that the supporting bias field depends e.g. on the exact width of the nanostructures. This tells us that our devices are still non-optimized for reporting the minimum*

power for magnon-induced switching. Still we highlight that in the pioneering experiment on electrically driven STT in “equally” non-optimized nanopillars (which we cited in the original manuscript) the (large) electrical power for reversal of 6 microWatts was extracted at a finite field (and not at zero field). Thereby we considered a similar experimental setting for the comparison mentioned in the manuscript. Note that our power levels are already nanoWatts, i.e., three orders of magnitude below the pioneering work on STT! We expect to reach power levels below nW once the microscopic switching is understood. We agree that at the current stage we use a magnetic field to break the symmetry for magnetic reversal. Considering Refs. [] on magnon-mediated reversal without symmetry-breaking field, future devices might be optimized in such a way that the bias field is obsolete.

“Finally, I could not find the results about the switching of the FIM (YIG) as the author claim as one of their main contributions.”

Response: We have not discussed the switching of YIG. YIG is a soft magnet with a very small coercive field. The technology-relevant switching phenomenon occurs in the (hard-magnetic) permalloy stripes which are considered as the magnetic bits.

Brief List of Changes:

We have rephrased parts of

- the abstract,
- main text and
- figure captions

without changing the contents and conclusions drawn (The figures have not been altered).

We have extended the discussion on MAMR and on the microscopic origin(s) of the observed magnon-induced reversal.

We updated the list of references concerning the current state of the art.

Reviewers' Comments:

Reviewer #1:

Remarks to the Author:

In author's response letter, heating effects and crosstalk problem are well explained, and the part of the paper is properly revised. However, the authors' main claim that the device works at low power is still unsubstantiated. I want hear authors' answer to the following question [1] before I decide my recommendation

[1] p5 Line156 "Hence, the RF-field-excited SWs in YIG reduce the stripes' switching fields about 30 times more efficiently than a static Oersted field applied oppositely to MPy." :
What is the basis for "30 times"? (According to the paper, the change in the threshold value for the magnetization reversal looks at most 30% when the amplitude of the input microwave magnetic field is changed by a factor of 7.)

(minor comments)

[2] p8 Line 237 "In Ref. [42], a directly injected electrical current was used to switch one single Co nanomagnet of a small volume ($60 \times 130 \times 2.5$) nm³. A minimum power $P_{el} \sim 6$ microW was required. In our study, we evaluated a critical power of $P_{C1,prec} = 3.4$ nW when reversing a few 10 Py nanostripes of a much larger volume ($100 \times 26000 \times 20$) nm³ each.":

Why did the authors conclude that the power required for the reversal is 3.4 nW? (The microwave input power in Fig. 1 (f) is about 10microW, which is equivalent to the value of 6microW in the cited paper.)

Reviewer #3:

Remarks to the Author:

I have read the response of the authors. Unfortunately, I still find this study does not make significant progress, and I cannot recommend publishing this manuscript in Nature communications. Still the authors did not provide enough evidences for the reversal of the magnetization only by spin waves as also mentioned by other reviewers. The physical understanding is not clear and fully discussed in this manuscript. I can not find the progressiveness and universality of their proposals. Since the devices with field-assisted is not optimized as they said, I am concerned about the estimation of the power consumptions compared with STT.

Detailed Responses

Reviewer #1 (Remarks to the Author):

In author's response letter, heating effects and crosstalk problem are well explained, and the part of the paper is properly revised. However, the authors' main claim that the device works at low power is still unsubstantiated. I want hear authors' answer to the following question [1] before I decide my recommendation

[1] p5 Line156 "Hence, the RF-field-excited SWs in YIG reduce the stripes' switching fields about 30 times more efficiently than a static Oersted field applied oppositely to MPy."

What is the basis for "30 times"? (According to the paper, the change in the threshold value for the magnetization reversal looks at most 30% when the amplitude of the input microwave magnetic field is changed by a factor of 7.)

Response: We have rephrased the paragraph. We now write:

In Fig. 3 b we plot the experimentally extracted critical fields H_{C1} and H_{C2} (blue and orange symbols, respectively) versus the rms-value of the dynamical magnetic field component $\mu_0 h_{rf,x}$ of the radiofrequency field (RF) generated by the CPW. The amplitude is calculated via $\mu_0 h_{rf,x} = \mu_0 \sqrt{P_{irr}/(2Z_0 w_L^2)}$ [33] ($Z_0 = 50 \Omega$ and $w_L = 2.1 \mu\text{m}$, see Supplementary Fig. 1). The two dashed lines with negative slopes show that the nanostripes switch at smaller applied fields when $h_{rf,x}$ is increased. The top dashed line starts at values $(\mu_0 h_{rf,x}, H_{C2}) = (x_1, y_1) = (0.06 \text{ mT}, 34 \text{ mT})$ and ends at $(x_2, y_2) = (0.34 \text{ mT}, 25 \text{ mT})$. Hence, an increase in RF amplitude of $\Delta(\mu_0 h_{rf,x}) = x_2 - x_1 = (0.34 - 0.06) \text{ mT} = 0.28 \text{ mT}$ leads to a switching field reduction by $\Delta(H_{C2}) = -(y_2 - y_1) = (34 - 25) \text{ mT} = 9 \text{ mT}$. The ratio $\Delta(H_{C2})/\Delta(\mu_0 h_{rf,x})$ amounts to $9 \text{ mT}/0.28 \text{ mT} = 32.1$, that is, a small (positive) difference in $\mu_0 h_{rf,x}$ enables a 30 times larger (negative) difference in H_{C2} .

(minor comments)

[2] p8 Line 237 "In Ref. [42], a directly injected electrical current was used to switch one single Co nanomagnet of a small volume ($60 \times 130 \times 2.5$) nm³. A minimum power $P_{el} \sim 6 \mu\text{W}$ was required. In our study, we evaluated a critical power of $P_{C1,prec} = 3.4 \text{ nW}$ when reversing a few 10 Py nanostripes of a much larger volume ($100 \times 26000 \times 20$) nm³ each."

Why did the authors conclude that the power required for the reversal is 3.4 nW? (The microwave input power in Fig. 1 (f) is about 10 microW, which is equivalent to the value of 6 microW in the cited paper.)

Response: We thank the referee for this comment. In the pioneering work on spin-transfer torque (STT) induced switching a charge current I was applied to the nanopillar in the out-of-plane direction. All the electrons which form the spin-polarized current I go through the nanomagnet and follow the same potential difference V across the nanopillar. Accordingly one needs to provide the electrical power $P = IV$ for the switching. This power P is fully "consumed(absorbed)" in the nanopillar. In the pioneering work, P amounted to $6 \mu\text{W}$.

For the spin waves, the power consideration is different. The power P_{irr} of the radiofrequency (RF) signal applied to the CPW is reflected to a very large extent (if the CPW forms e.g. a short). The absorbed power P_{prec} for the relevant microwave-to-magnon transduction is very small. We showed in the manuscript how to calculate P_{prec} from the applied power P_{irr} :

$P_{prec} = P_{irr} \cdot [\text{Mag}(S_{11})]^2$. The measured data $\text{Mag}(S_{11})$ are shown in Fig. 1h and 1i. For the mode labelled by 1G we find $\text{Mag}(S_{11}) = 7 \cdot 10^{-3}$. Hence, for this mode the "absorbed" power is $P_{irr} \cdot (0.007)^2 = 68.9 \mu\text{W} \cdot 0.000049 = 0.0034 \mu\text{W} = 3.4 \text{ nW}$. Many researchers in the magnonics community work on approaches to significantly reduce P_{irr} for the same P_{prec} . Please note however that the large amount of reflected RF power is not "lost". By means of a circulator it can be routed to the next CPW integrated to a further magnonic circuit and so on. This is different from the

STT nanopillar in which all the power is used in the nanopillar. **We added corresponding comments to the manuscript.**

Reviewer #3 (Remarks to the Author):

I have read the response of the authors. Unfortunately, I still find this study does not make significant progress, and I cannot recommend publishing this manuscript in Nature communications. Still the authors did not provide enough evidences for the reversal of the magnetization only by spin waves as also mentioned by other reviewers.

Response: We are happy to provide the statement extracted from the report of reviewer 1: “In author’s response letter, heating effects and crosstalk problem are well explained, and the part of the paper is properly revised.” The cross-talk is -50 dB, and so small that the nanostructures which are 25 μm away from the emitter are not directly affected by the RF signal. We attribute the observed reversal to the spin waves launched at the emitter CPW.

The physical understanding is not clear and fully discussed in this manuscript.

Response: We have discussed possible mechanisms. Based on a series of samples which we have investigated we mentioned the most likely microscopic origin. This is the torque provided by the spin waves’ stray field (dipolar effect).

I can not find the progressiveness and universality of their proposals.

Response: If theoreticians confirm that a dipolar effect (see above) switches the nanostructures, this mechanism is “universal”. It would mean that technologically the interface properties between the ferrimagnet and the ferromagnet do not play a role. Sputtering a ferromagnet on top of YIG would be enough to realize the corresponding device. This is attractive in view of a technically easy and cost-effective deposition technology which is magnetron sputtering.

Since the devices with field-assisted is not optimized as they said, I am concerned about the estimation of the power consumptions compared with STT.

Response: The power consumption of the spin system is calculated based on the power P_{prec} absorbed by the spin system as outlined in detail above. This is **not** an estimation. We consider the absolute values known from the experimental parameters and data. The electrical power for the pioneering STT device is also given in absolute numbers. The latter device was not optimized either. For technologically relevant STT-RAMs the device designs were then made more efficient. We expect the same optimization to happen for the phenomenon reported in our present manuscript. We expect that the required spin-precessional power P_{prec} will be further reduced below 3.4 nW as soon as the microscopic origin is identified by theoreticians. They might even propose a hybrid system which combines efficiently dipolar effects, spin pumping and magnon torque. At the same time the nanomagnets can be optimized concerning the most efficient reversal mechanism. Such considerations about device optimization motivate us to compare our spin-wave-induced reversal experiment with the pioneering STT-induced reversal experiment.

Brief list of changes:

We shortened the abstract to 149 words

We added a discussion explaining the factor of 30

We added sentences to explain the difference between input power and spin-precessional power.